# Antioxidant and Anticancer Activity of *Vitis vinifera* Extracts in Breast Cell Lines

**DOI:** 10.3390/life14020228

**Published:** 2024-02-06

**Authors:** Evgenia Maria Tsantila, Nils Esslinger, Maria Christou, Panagiotis Papageorgis, Christiana M. Neophytou

**Affiliations:** 1Apoptosis and Cancer Chemoresistance Laboratory, Basic and Translational Cancer Research Center, Department of Life Sciences, European University Cyprus, Nicosia 2404, Cyprus; et221233@students.euc.ac.cy; 2Department of Research and Development, Alpinamed AG, 9306 Freidorf, Switzerland; nils.esslinger@alpinamed.ch; 3Tumor Microenvironment, Metastasis and Experimental Therapeutics Laboratory, Basic and Translational Cancer Research Center, Department of Life Sciences, European University Cyprus, Nicosia 2404, Cyprus; mc211170@students.euc.ac.cy (M.C.); p.papageorgis@euc.ac.cy (P.P.)

**Keywords:** *Vitis vinifera*, antioxidants, chemoprevention, natural anticancer agents, white grape, polyphenols, breast cancer, white grape extracts, dry seeds extracts, reactive oxygen species

## Abstract

*Vitis vinifera* extracts have been shown to possess antioxidant activity because of their polyphenol content. In addition, their therapeutic potential against several diseases, including cancer, has been reported. In this study, we produced twelve extracts from the seeds, fruit, leaves, and wood of the *Vitis vinifera* Airen variety using different extraction methodologies and measured their total polyphenol content (TPC). We also determined their antioxidant and antiproliferative effects against normal cells and evaluated the most potent extract against a panel of breast cancer cell lines. We found that the extracts produced by the seeds of *Vitis vinifera* had a higher TPC compared to the other parts of the plant. Most extracts produced from seeds had antioxidant activity and did not show cytotoxicity against normal breast cells. The extract produced from whole organic seeds of white grape showed the best correlation between the dose and the ROS inhibition at all time points compared to the other seed extracts and also had antiproliferative properties in estrogen-receptor-positive MCF-7 breast cancer cells. Its mechanism of action involves inhibition of proteins Bcl-2, Bcl-xL, and survivin, and induction of apoptosis. Further investigation of the constituents and activity of *Vitis vinifera* extracts may reveal potential pharmacological applications of this plant.

## 1. Introduction

Breast cancer remains the first in incidence and second in mortality rate malignancy in women [1]. Even though novel therapeutic agents have significantly improved the outcome of this disease, more research is needed to discover compounds that both reduce the risk for cancer development but can also target cancer cells. Natural extracts have long been used as a source for agents with chemopreventive and chemotherapeutic properties [2]. The use of natural supplements in cancer aims to reduce drug resistance, acting synergistically with anticancer drugs to lower drug concentrations and, therefore, diminish the adverse side effects associated with cancer therapy.

Various pharmacological properties of extracts derived from grape *Vitis vinifera* (family Vitaceae) have been reported in recent years [3]. Extracts produced from the leaves, fruit (skin and seeds), and other parts of this plant include antioxidant, anticancer, antibacterial, and antidiabetic activities and have displayed cardioprotective, hepatoprotective, and neuroprotective effects [4,5,6,7,8,9,10]. Grape-derived compounds and products can block metabolic syndrome in vitro and in vivo and improve heart conditions [11,12]. The therapeutic effects of *Vitis vinifera* are attributed to the active constituents of its parts, mostly to the presence of polyphenolic compounds [13]. The phenolic compounds found in grape include anthocyanins, flavanols, flavonols, stilbenes, and phenolic acids [14,15,16]. Resveratrol, a stilbene found in grapes and its byproducts, has been widely studied for its antioxidant and anticancer properties [17]. Flavonoids, mostly distributed in grapes and especially in seeds, principally contain catechins, epicatechin, and procyanidin polymers [18]. *V. vinifera* seeds contain considerable quantities of gallic acid and p-coumaric, in greater amounts than the seeds of other fruit varieties [19]. 

Mixtures produced from *Vitis vinifera* peels and root contain high amounts of phenolics and flavonoids and display excellent antioxidant capacity [20,21]. Seeds extracted from different *Vitis vinifera* varieties have high TPC and antioxidant activity; leaf extracts also show antioxidant activity both in vitro and in vivo [10,22]. Importantly, recent evidence suggest that extracts and isolated compounds from *Vitis vinifera* elicit anticancer activity via distinct underlying mechanisms including activation of the immune system in colon cancer cells, suppressing metastasis in aggressive breast cancer via downregulation of interleukin 1 alpha (IL-1α), and affecting gene expression of fatty acid-binding protein 5 (FABP5) to block the proliferation of prostate cancer cells [23,24,25].

The antioxidant profile of the extracts depends not only on the source of the plant material (fruit, seeds, leaves) but also on the extraction methodology [26]. Certain studies report that the concentration of the extract used may have either a pro-oxidant or antioxidant effect in vitro or in vivo [27]. In this study, we examined the chemopreventive and anticancer potential of twelve *Vitis vinifera* extracts (E1–E12) produced from various parts of the plant using several extraction methods and solvents. We found that the extracts produced from the seeds of the *Vitis vinifera* (Airen variety) had higher TPC compared to extracts from the leaves, fruit (without seeds), and wood. Most extracts had antioxidant activity and reduced endogenous ROS production in MCF-10A normal breast cells, without displaying significant cytotoxicity. The extract derived from organic seeds from white grape also showed selective toxicity against estrogen-receptor-positive (ER^+^) MCF-7 breast cancer cells, reducing proliferation and inducing apoptosis via lowering the levels of antiapoptotic proteins Bcl-2, Bcl-xL, and survivin. The extract was not effective in triple negative (ER^−^, PR^−^, HER2neu^−^) MDA-MB-231-LM2 breast cancer cells, suggesting that its mechanism of action may involve the ER pathway. Further insights into the mode of action of *Vitis vinifera* extracts in breast cancer may reveal potential uses for therapeutic and chemopreventive approaches.

## 2. Materials and Methods

### 2.1. Plant Material

Extracts E1–E4 were produced from *Vitis vinifera* (mixed white and red grape) from traditional agriculture, origin unknown. For the seed production process, the combustion air waste heat from the vine wood and dried pomace was conducted at an initial temperature of approx. 120–150 °C through an external channel separated from the fresh pomace. The fresh pomace was gently dried at a temperature of approximately 60–70 °C, over a distance of approx. 20 m for 15–20 min from 60% moisture to a residual moisture of less than 10%. In the same continuous process, the grape seeds were selected from the fresh dried grape pomace by rotary drum sieve (pre-selection) and fine separation by vibrating screen.

Extracts E5–E12 were produced from *Vitis vinifera* (white grape, Airen variety) from organic agriculture, originated from western Europe. 

For the seed production process, the combustion air waste heat from the organic vine wood (used for extract E12) and dried pomace (used for extracts E8–E10) was conducted at an initial temperature of approx. 120–150 °C through an external channel separated from the fresh pomace. The fresh pomace was gently dried at a temperature of approximately 60–70 °C, over a distance of approx. 20 m for 15–20 min from 60% moisture to a residual moisture of less than 10%. In the same continuous process, the grape seeds were selected from the fresh dried grape pomace by rotary drum sieve (pre-selection) and fine separation by vibrating screen. The seeds were additionally cleaned by wind screening to increase their purity.

### 2.2. Extraction Methodology

All of the extracts were made by maceration; therefore, the plant material was put into the solvent under stirring. After 15 min of stirring the mixture was well closed and macerated for 24 h; after that, the plant material was separated by filtration through a glass fiber filter (1 µm). The plant material to solvent ratio (m/m) was the same for all extracts. The variations were mainly in the solvent used for extraction (water E1, ethanol 70%m/m E2, and ethanol 62%m/m E3–E12). The other variations were the pre-washing step of the plant material with water (E5 versus E6 and E7; E8 versus E9 and E10). All variations are described in Table 1.

Extract E1 was produced with *Vitis vinifera* seeds from traditional agriculture. Therefore, the seeds were extracted with a mixture of ethanol and purified water, the resulting liquid was distilled under vacuum and dried under vacuum to a moisture of less than 5%.

Extract E2 was produced with *Vitis vinifera* seeds from traditional agriculture. Therefore, the seeds were extracted with purified water, the resulting liquid was distilled under vacuum and dried under vacuum to a moisture of less than 5%.

Extracts E3 and E4 were produced with the pomace of the oil-pressing of *Vitis vinifera* seeds from traditional agriculture. The pomace was extracted with a mixture of ethanol and purified water. The resulting liquid was distilled under vacuum to 20% of the initial mass. The precipitation was separated and dried under vacuum (E3), the separated liquid was distilled under vacuum and dried under vacuum (E4).

Extract E5 was produced with *Vitis vinifera* seeds from organic agriculture. Therefore, the seeds were extracted with a mixture of ethanol and purified water, the resulting liquid was distilled under vacuum and dried under vacuum to a moisture of less than 5%.

Extracts E6 and E7 were produced with *Vitis vinifera* seeds from organic agriculture. Therefore, the seeds were pre-washed with purified water, the resulting liquid was distilled under vacuum and dried under vacuum (E6). The pre-washed seeds were extracted with a mixture of ethanol and purified water, the resulting liquid was distilled under vacuum and dried under vacuum to a moisture of less than 5% (E7).

Extract E8 was produced with dried *Vitis vinifera* pomace without seeds from organic agriculture. Therefore, the pomace was extracted with a mixture of ethanol and purified water, the resulting liquid was distilled under vacuum and dried under vacuum to a moisture of less than 5%.

Extracts E9 and E10 were produced with dried *Vitis vinifera* pomace without seeds from organic agriculture. Therefore, the pomace was prewashed with purified water, the resulting liquid was distilled under vacuum and dried under vacuum (E9). The pre-washed pomace was extracted with a mixture of ethanol und purified water, the resulting liquid was distilled under vacuum and dried under vacuum to a moisture of less than 5% (E10).

Extract E11 was produced with the leaves of *Vitis vinifera* from organic agriculture naturally dried without heating. The leaves were extracted with a mixture of ethanol and purified water, the resulting liquid was distilled under vacuum and dried under vacuum to a moisture of less than 5%.

Extract E12 was produced with the dried wood of *Vitis vinifera* from organic agriculture. The wood was extracted with a mixture of ethanol and purified water, the resulting liquid was distilled under vacuum and dried under vacuum to a moisture of less than 5%.

### 2.3. Total Polyphenol Content

The measurement of the total polyphenol content was based on the general European Pharmacopeia method 20814 “Tannins in herbal drugs (2.8.14.)”, Section 1 total polyphenols. As reference substances, pyrogallol and catechin were used to prepare for each substance three standard solutions; the method was adapted for automation of the photometric measurement using a HPLC Device (Dionex Ultimate 3000, Dionex Softron GmbH, Germering, Germany). The samples and reference substances were dissolved in purified water by using an ultrasonic bath, the solution was filtrated through a 0.2 µm syringe filter (regenerated cellulose) into 1.5 mL HPLC-vials. A total of 20 µL of the standard or sample solution was pipetted by the autosampler into a reaction vial, 10 µL of the Folin and Ciocalteu’s phenol reagent (2 M with respect to acid) was pipetted by the autosampler into the reaction vial and mixed, 100 µL purified water was pipetted by the autosampler into the reaction vial and mixed, and at least 249 µL sodium carbonate decahydrate solution 290 g/L was pipetted by the autosampler into the reaction vial and mixed. After 30 min, the autosampler injected 10 µL out of the reaction vial. As eluent, purified water was used. The absorption was measured at 760 nm with 4 nm wide band. Instead of an analytical column, a restrictor capillary was used. All reagents and reference substances were purchased from Sigma Aldrich (St. Louis, MO, USA).

A calibration curve was generated using the concentrations of the standard solutions (mg/mL) and the resulting areas of the peak.

The concentration of the samples in % (m/m) were calculated using following formula:Concentration Sample (%) = Concentration (mg/mL) × 100 × Volume of Sample Solution (mL)/Sample weight (mg)

### 2.4. Oligomeric Proanthocyanidins (OPC) Content

The measurement of the oligomeric proanthocyanidins content were based on the literature [28]. With this method one of the main components was characterized. The sum of monomers (catechin, epicatechin etc.) and each of the polymer groups from DP2 up to DP8 were quantified separately. The samples and reference substance catechin were dissolved in methanol 70%*v*/*v* by using an ultrasonic bath, the solution was filtrated through a 0.2 µm syringe filter (regenerated cellulose) into 1.5 mL HPLC vials. The HPLC conditions were as follows: column temperature: 35 °C; flow: 1 mL/min; eluent A: 98% acetonitrile, 2% acetic acid; eluent B: 95% methanol, 3% water, 2% acetic acid; gradient: 7%B for 3 min, to 30%B in 12 min, to 49%B in 25 min. The OPC groups were differentiate by retention time windows: monomers 1.1–2.0 min, DP2 2.55–3.45 min, DP3 5.25–6.75 min, DP4 8.75–10.25 min, DP5 11.62–12.88 min, DP6 14.0–15.0 min, DP7 15.95–16.85 min, DP8 17.65–18.55 min. For DP2, the suitable reference substances procyanidin A1 and procyanidin B1 were used to verify the retention time window, for DP3 the suitable reference substance procyanidin C1 was used to verify the retention time window.

The chromatography was executed by HPLC with a Luna 5 µm HILIC 200 Å 150 × 3 mm (Phenomenex) column. The fluorescence was measured with an excitation wavelength of 231 nm and an emission wavelength of 320 nm. All reagents and reference substances were purchased from Sigma Aldrich.

### 2.5. Cell Culture and Reagents

MCF-7 and MCF-10A cell lines were obtained from the American Type Culture Collection (ATCC) (Manassas, VA, USA). The MDA-MB-231-LM2 cell line was derived by Joan Massagué’s group from MDA-MB-231 cells [29]. MCF-7/TAM-R cells were a kind gift from Dr. I. Hutcheson (Tenovus Centre for Cancer Research, Cardiff University). HUVEC cells were a kind gift from Dr Cristina Fornaguera (IQS School of Engineering Universitat Ramon Llull, Barcelona, Spain. MCF-7, MCF-7/TAM-R, and MDA-MB-231-LM2 breast cancer cell lines were cultured in DMEM supplemented with 10% fetal bovine serum (FBS) and 1% antibiotic/antimycotic. MCF-10A immortalized breast cell line was cultured in DMEM F12 supplemented with 20 ng/mL EGF, 100 ng/mL cholera toxin, 500 ng/mL hydrocortisone, 10 ng/mL insulin, 5% horse serum (HS), and 1% antibiotic/antimycotic. DMEM, FBS, HS, antibiotic/antimycotic, and trypsin were purchased from Gibco, Invitrogen (Carlsbad, CA, USA). HUVEC cells were cultured in endothelial cell growth media (Sigma Aldrich). Bcl-2, Bcl-xL, survivin, and GAPDH antibodies were purchased from Cell Signaling Technology (Danvers, MA, USA). *Vitis vinifera* extracts and polyphenolic standards were provided by Alpinamed AG (Freidorf, Switzerland). All other reagents were purchased from Sigma Aldrich.

### 2.6. Extract Dilution

Extracts were diluted in sterile PBS at 1 mg/mL stock, filtered through 0.2 μm pore filter, and stored at −20 °C for up to 3 weeks. All samples are diluted to 10%, 50%, and 100% bioavailability. For the 100% bioavailability, we considered a maximum intake of 225 mg daily for an average of 6.0 L of total volume of blood. The concentrations used for the in vitro studies were the following:100% bioavailability = 225 mg/6.0 L = 37.5 mg/L
50% bioavailability = 37.5 mg/L × 0.5 = 18.75 mg/L
10% bioavailability = 37.5 mg/L × 0.1 = 3.75 mg/L

### 2.7. MTT Assay

A total of 5 × 10^4^ cells were seeded per well of a 96-well plate. The breast cell lines MCF-7, MCF-10A, MDA-MB-231-LM2, and endothelial HUVEC cells were incubated overnight to allow for cell attachment and recovery. Cells were treated with increasing doses of *Vitis vinifera* extracts (3.75–37.5 mg/L) and incubated for 24–72 h at 37 °C. Cell viability was measured using the MTT 3-(4,5-dimethylthiazol-2-yl)-2,5-monotetrazolium bromide assay [30]. At the end of each incubation period, 20 μL of MTT dye (1 mg/mL; Sigma, St. Louis, MO, USA) was added in each well and the plate was incubated at 37 °C for 4 h. Media was removed and 200 μL of DMSO was added. Subsequently, the plates were placed on a shaker for 15 min and read on a microplate reader (Varioskan, Thermo Fisher Scientific, Waltham, MA, USA) at 570 nm. Absorbance was proportional to the number of viable cells per well. Percentage of cell viability in each group was calculated after normalization to its own control (PBS added at the same concentration as treatments).

### 2.8. DCFH-DA Assay

2′,7′-Dichlorofluorescein diacetate (DCFH-DA) is a lipophilic non-fluorescent cell-permeable redox probe. The DCFH-DA readily crosses the cell membrane through passive diffusion followed by deacetylation. The deacylated product is an oxidant sensitive 2′,7′-dichlorofluorescein (DCHF). DCHF is oxidized later to form highly fluorescent DCF, which is measured at excitation 485 nm/emission 535 nm. To measure inhibition of endogenous ROS, cells were plated in 96-well plates (4 × 10^5^ cells/well) and left to attach for 24 h. Next, extracts at 3.75, 18.75, and 37.5 mg/L were added with 1 μL of DCFH-DA (100 μΜ) in each well and plates were covered and left to incubate at 37 °C for 1 h. 

Media without extracts was used as negative control while media with DCFH-DA only was used as positive control. Media was removed and cells were washed ×2 with 1× PBS. At the end of the wash, 100 μL of 1× PBS was added in each well and fluorescence was measured by a plate reader at exc. 485 nm/ em. 535 nm at 0, 10, 20, and 60 min (Appendix A). The effect of PBS (extract diluent) in ROS inhibition was also measured at the same concentrations as the extracts.

To calculate the cellular antioxidant activity (CAA), we first measured the emission at 535 nm for each extract concentration and corresponding PBS volumes per time and calculated the area under curve (AUC). The CAA values for each concentration of extract were calculated as follows:CAA Units = 100 − (AUC_Extract_/AUC_PBS_) × 100

The effective dose (CAA50) was determined for each extract from the concentration (mg/L) plot versus CAA units (Appendix A).

### 2.9. Cell Cycle Analysis

Cells were treated with different concentrations of extracts as described in the figure legends. Following incubation, cells were harvested, fixed in 70% ethanol, incubated with the propidium iodide (PI) staining solution (containing 1 mg/mL PI and 100 μg/mL Rnase) for 30 min at 37 °C, and analyzed for DNA content using the Attune NxT flow cytometer (Thermo Fisher Scientific, MA, USA) and the FlowJo analysis software V10.10.0 (BD Biosciences, Franklin Lakes, NJ, USA).

### 2.10. Annexin V/Propidium Iodide Staining

Cells were seeded at a concentration of 1 × 10^5^ cells per well of a 60-mm plate and treated with the extracts as indicated. Cells were harvested and stained as described by Alexa Fluor^TM^ 488 Annexin V/Dead Cell Apoptosis kit (Life Technologies, Carlsbad, CA, USA). Cell viability, death, and apoptosis were evaluated using the Attune NxT flow cytometer (Thermo Fisher Scientific, MA, USA) and the FlowJo analysis software (BD Biosciences, NJ, US). The annexin-V-positive/PI-negative cells were recognized as early apoptotic cells by the FlowJo analysis software V10.10.0 (BD Biosciences, Franklin Lakes, NJ, USA).whereas the annexin-V-positive/PI-positive cells were identified as late apoptotic/dead cells. Similarly, the annexin-V-negative/PI-negative cells were identified as viable cells.

### 2.11. Total RNA Preparation and Real-Time Quantitative PCR (q-PCR)

Total RNA was extracted with Trizol reagent (Invitrogen, Carlsbad, CA, USA) following the manufacturer’s protocol. cDNA was synthesized with random primers using the Superscript III Reverse Transcriptase (Invitrogen, Carlsbad, CA, USA). Primer sequences were designed using Primer3 and are as follows: human Bcl-2, 5′-ATGTGTGTGGAGAGCGTCAA-3′ (forward) and 5′-ACAGTTCCACAAAGGCATCC-3′ (reverse), human Bcl-xL, 5′-GTAAACTGGGGTCGCATTGT-3′ (forward) and 5′-TGGATCCAAGGCTCTAGGTG-3′ (reverse), human ER-beta, reverse 5′-TCAGGCATGCGAGTAACAAG-3′ (reverse), forward 5′-CTCCAGCAGCAGGTCATACA-3′ (forward), human survivin, 5′-GACGACCCCATAGAGGAACA-3′ (forward) and 5′-GACAGAAAGGAAAGCGCAAC-3′ (reverse); and human GAPDH, 5′-TTGGTATCGTGGAAGGACTCA-3′ (forward), 5′-TGTCATCATATTTGGCAGGTTT-3′ (reverse). Real-time PCR was performed using the BioRad CFX96 Real-Time System and the SYBR Green PCR Master Mix (Applied Biosystems, Waltham, MA, USA) according to the manufacturer’s instructions. The PCR products were normalized to those obtained from GAPDH mRNA amplification and gene expression was quantified using the ΔΔCt method [31].

### 2.12. Western Blot

Following incubation with selected extracts at different concentrations (as indicated in the figures), cells were washed with ice-cold PBS and lysed in RIPA buffer (150 mM NaCl, 50 mM Tris, 5 mM EDTA [Na2], 1% (*v*/*v*) Triton X-100, 1% (*w*/*v*) deoxycholate (24 mM), 0.1% (*w*/*v*) SDS (35 mM)) containing protease and phosphatase inhibitors (Complete Mini, Roche, Basel, Switzerland), in order to achieve the cleavage of the cell membranes. The total cellular extracts of the proteins were collected and the protein levels in each sample were measured by using the Bradford method and run on SDS page electrophoresis as described elsewhere [32]. Membranes were incubated using SuperSignal West FemtoSubstrate (Thermo Scientific, Waltham, MA, USA) per the manufacturer’s instructions and visualized using the BioRad Universal Hood II and the Image Lab 5.0 software. The intensity values from the densitometry analysis of Western blots were normalized against the corresponding loading control using ImageJ software analysis v1.53e (NIH).

### 2.13. Statistical Analysis

Results for continuous variables were presented as mean standard deviation. Two-group differences in continuous variables were assessed by the unpaired *t*-test. *p*-values are two-tailed with confidence intervals 95%. Statistical analysis was performed by comparing treated samples with untreated control. All statistical tests were conducted using Prism software version 8.0 (GraphPad, San Diego, CA, USA).

## 3. Results

### 3.1. Composition of Vitis vinifera Extracts

We investigated the TPC in the produced extracts expressed as pyrogallol and catechin. We found that the extracts produced from seeds by different extraction methods, had significantly higher polyphenolic content compared to the extracts produced by fruits without seeds, leaves, or wood (Table 1). TPC expressed as catechin was in higher amounts (ranging from 27.53–38.92 g/100 g) in the extracts produced from seeds compared to pyrogallol (ranging from 20.44–28.89 g/100 g). Overall, the whole seeds dry-extract produced with ethanol 70%m/m (E1) had the highest TPC measured both as pyrogallol or catechin, followed by the dry-extract produced from crushed seeds without oil produced with ethanol 62%m/m (E4). In the case of the low solubility of the dry-extract E3 in water, the TPC measurement was also made with methanol as sample solvent for this extract (TPC of E3 as pyrogallol in H_2_O 12.44 g/100 g versus TPC of E3 as pyrogallol in methanol 33.27%). The order of the produced extracts based on their TPC was as follows: E1 > E4 > E7 > E2 > E5 (> E3) > E6 > E10 > E8 > E11 > E9 > E12.

We investigated also the oligomeric proanthocyanidins content (OPC) in the seed extracts. The results were grouped by the degree of polymerization (DP) into the following groups: catechins (DP1 = monomers) OPC DP 2–4, OPC DP 5–8, OPC DP 1–8. The order of the produced seed dry extracts based on their OPC DP1–8 was as follows:E4 > E1 > E7 > E2 > E5 > E3 > E6

### 3.2. Effect of Vitis vinifera Extracts on the Viability of Normal Breast Cells

Initially, we investigated the cytotoxic effects of *Vitis vinifera* extracts (E1–E12) against normal MCF-10A “immortalized” breast cell lines. Cells were treated with increasing concentrations of extracts for 72 h (Figure 1). Concentrations were chosen based on the 10%, 50%, and 100% bioavailability that can be reached in vivo as described in Section 2 We observed, with the exception of E3 and E5, that the *Vitis vinifera* extracts were not cytotoxic and did not inhibit the proliferation of normal MCF-10A cells (Figure 1). The concentration needed to inhibit the proliferation of cells up to 50% was calculated only for E3 (38.49 mg/L) and E5 (131.8 mg/L). Extracts E3 and E5 that were found to be cytotoxic against normal immortalized cells were not evaluated further for their potential chemopreventive activity.

### 3.3. Antioxidant Activity of Vitis vinifera Extracts

We evaluated the antioxidant capacity of *Vitis vinifera* extracts as described in the Methods section. Cells were incubated with increasing concentrations of the extracts for 1 h in the presence of DCFH-DA; free radicals present within the cell convert DCFH-DA to highly fluorescent DCF [33]. Any reduction in the fluorescent signal at 535 nm indicates quenching of free radicals by the extracts. We measured the ROS inhibition achieved by increasing concentration (3.75, 18.75, 37.5 mg/L) of extracts vs. time (Figure 2A) and plotted the reduction in ROS vs. concentration at 60 min (Figure 2B). E7, E8, and E12 show a dose-depended inhibition of ROS (Figure 2A). In the case of E8, reduction in fluorescence was observed at 18.75 and 37.5 mg/L only. Based on the slope of each graph (linear regression), shown in Figure 2B, the antioxidant potency of the examined extracts is evaluated as follows: E12 > E8 > E7 > E11 > E1 > E2 > E4 > E10 > E6 > E9. Extracts E6, E9, and E10 have a positive slope in Figure 2B, indicating that they had no ROS inhibitory activity after 60 min of incubation.

We further calculated the CAA50 value for each extract (Table 2), based on the graph (concentration of extract vs. CAA units) created for each extract (Appendix A), as described in the Methods section. E1 can achieve the CAA50 at the lowest concentration (31 mg/L), followed by E4, E2, E7, E11, and E12. For extracts E6, E9, and E10, the concentration needed to reach CAA50 was not evaluable.

We chose to investigate E7 further for anticancer activity because it showed the best correlation between the dose and the ROS inhibition at all time points compared to the other seed extracts (Figure 2A,B).

### 3.4. Investigation of the Anticancer Activity of Whole Seeds Extract from Vitis vinifera in Breast Cancer Cells

To further examine the antiproliferative activity of E7, we tested the *Vitis vinifera* seeds extract against MCF-7 (ER+, PR+) and MDA-MB-231-LM2 (ER−, PR−, HER2neu−) breast cancer cells. We found that E7 was able to reduce the viability of MCF-7 cells at all concentrations tested. Its IC50 was calculated at 46.73 mg/L at 72 h (Figure 3A). MDA-MB-231-LM2 triple-negative cells were unaffected by the treatment. To investigate whether the presence of the estrogen receptor (ER) may play a role in the efficacy of *Vitis vinifera* seeds extract, we examined its antiproliferative action in normal endothelial HUVEC cells that express ER-beta as well as in the MCF-7-derivative MCF7-TAMR cell line, where the ER has been silenced. We found that HUVEC cells expressing the ER were sensitive to E7 treatment (IC50 53.67 mg/L), while the MCF-7 derivative cell line, MCF-7 TAMR, was not affected by E7 at 72 h of treatment (Figure 3A). The mRNA levels of ER-beta for all cell lines were confirmed by real-time PCR assay (Figure 3B).

To further elucidate the mechanism of action of E7 in breast cancer cells, we performed cell cycle analysis. We show that following incubation at 48 h, the highest concentration of E7 causes the appearance of the subG1 fraction (50.3% at 37.5 mg/L vs. 2.05% in PBS treatment) in MCF-7 cells, indicative of apoptosis (Figure 4A). Apoptosis induction was further confirmed by annexin V/PI staining; E7 increased the percentage of early apoptotic cells (14.9% at 37.5 mg/L vs. 5.46% in PBS treatment) in MCF-7 cells at 72 h treatment (Figure 4B). To investigate the underlying mechanism of apoptosis induction, we measured the changes in expression of important proteins implicated in cancer cell survival and apoptotic pathways. We found that E7 reduced the mRNA levels of antiapoptotic survivin, Bcl-xL, and Bcl-2 (Figure 4C), which was further confirmed by measuring the protein levels with Western Blot (Figure 4D). 

## 4. Discussion

*Vitis vinifera* extracts and isolated bioactive compounds have been previously reported to exhibit chemopreventive and antitumor activity. The pharmacological profile and bioactivity of *Vitis vinifera* extracts, especially from seeds, has been extensively studied; their antioxidant effects may be attributed to the presence of polyphenolic compounds, including (+)-catechin, (−)-epicatechin, flavanols, resveratrol, and proanthocyanidins [34,35]. Polyphenols are the most important phytochemicals in grape, with many reported biological activities and health-promoting benefits [18]. 

In this study, we investigated the polyphenol content in extracts produced by the European *Vinifera* cultivar. We used extracts from mixtures of red and white grape (E1–E4) as well as of white grape alone (E5–E12). It has been reported that red grapes contain higher TPC compared to white grapes [36,37]; however, the levels of polyphenols are depend on geoclimatic conditions and grape variety [38]. We show that the extracts produced by the seeds of the *Vitis vinifera*, using EtOH as a solvent, had much higher TPC content (measured as pyrogallol or catechin) compared to the leaves, wood, or fruit without seeds (Table 1). Even though grape seeds account for only 5% of the weight of the fruit, they contain 60–70% of the total polyphenols [22]. We also show that *Vitis vinifera* extracts produced from different parts of the plant, in their majority, did not affect the proliferation of normal MCF-10A cells (Figure 1) and have antioxidant properties (Figure 2, Table 2). To evaluate the potential selectivity of the grape extracts in normal vs. cancer cells, we chose to investigate antioxidant and antiproliferative activities using the same concentrations of extracts. Dietary polyphenols have chemopreventive activity by reducing the oxidative and inflammatory stress during the initiation and development of cancer; they can also counteract the side effects associated with drug therapy [39,40]. The juice of a red grape (*Vitis vinifera* L. cv. Aglianico N) variety not only demonstrated a direct radical-scavenging activity, but also counteracted doxorubicin-induced oxidative stress in normal cells, reducing ROS levels and suppressing caspase-3 activity [41,42].

The TPC and antioxidant capacity of plant extracts is also affected by the extraction solvent and methodology [43]. Overall, the whole seeds dry-extract produced with ethanol 70%m/m (E1) had the highest TPC measured both as pyrogallol or catechin and can achieve the CAA50 at the lowest concentration (Table 1, Table 2). E7 extracted with ethanol 62%m/m showed the best correlation between the dose and the ROS inhibition at all time points compared to the other seed extracts (Figure 2A,B). In a study aimed to optimize the extraction methodology in Isabella grape (*Vitis labrusca*), the resulting extract showed antioxidant capacity and cytotoxicity in MCF-7 breast cancer cells. The TPC expressed as milligrams of gallic acid equivalents per gram of sample (mg GAE/g sample) was 43.14 ± 5.00 mg [44]. Overall, aqueous methanol has proved an effective solvent for isolation of total phenolics and flavonoids from *Vitis vinifera* leaves as well as for increased radical scavenging and antioxidant activities [26].

Interestingly, we found that extracts E3 and E5 that had high TPC content (both measured in pyrogallol and catechin) were also cytotoxic against normal MCF-10A breast cells (Figure 1). However, there was no direct correlation between the TPC content and cytotoxicity, suggesting that other extracts constituents are responsible for their antiproliferative effects. In a thorough study of twenty-four *Vitis vinifera* grape cultivars, it was shown that the total antioxidant capacity of the cultivars was significantly correlated with the total phenolic and flavonoid content; however, no significant correlations were found between their antiproliferative effect and total phenolic or flavonoid content [5]. 

The potential anticancer activity of *Vitis vinifera* extracts has been previously documented. We show that E7, an extract produced by organic white grape seeds, was selectively cytotoxic in MCF-7 breast cancer cells and did not significantly affect the viability of normal MCF-10A breast cells (Figure 1 and Figure 3A). In addition, E7 had antioxidant capacity in MCF-10A cells (Figure 2A,B). The selectivity of natural extracts against cancer cells has been previously reported. A mixture of grape byproducts was protective in embryonic cardiomyocyte cells (H9c2) against oxidative damage produced by doxorubicin and less so in MCF-7 cells [42]. This may be attributed to the differences in the presence of cytoprotective enzymes as HO-1 and NQO1, responsible for the cellular detoxification of highly reactive molecules between normal and cancer cells [45]. Isolated compounds from *Vitis vinifera,* including stilbenes, have shown selective cytotoxicity against cancer cells but not normal human fibroblasts [46]. 

We showed that MCF-7 cells and HUVEC cells expressing the ER-beta were sensitive to treatment with E7, while ER-beta-negative cells MCF-7 TAMR and triple-negative MDA-MB-231 cells were not affected by the treatment (Figure 3A). The levels of ER-beta correlated with cell line sensitivity to treatment; the IC50 of E7 was lower in MCF-7 cells (46.73 mg/L) compared to HUVEC cells (53.67 mg/L), indicating that HUVEC cells were more resistant to treatment. The sensitivity to E7 correlated with the cells’ ER-beta mRNA expression levels (Figure 3B). Based on our results and the literature, MDA-MB-231-LM2 cells and MCF7 TAR cells do not express the ER-beta [29,47], while HUVEC and MCF-7 cells are positive for the expression of the receptor [48,49]. MCF-10A normal breast cells, which we found not being affected by E7 treatment (Figure 1), are also ER-beta-negative [50]. In a similar study investigating the cytotoxic effects of *Vitis Labrusca* extract on breast cancer cells, MCF-7 cells were also sensitive to treatment, whereas triple-negative MDA-MB-231 cells had no response to treatment [44]. Polyphenols, the main bioactive component of grape seeds extract, are known to bind to ERα and ERβ, and to exert properties that either mimic or antagonize the action of endogenous estrogens, even at low concentrations [51]. 

We also showed that E7 induced apoptosis in MCF-7 cells (Figure 4A,B). Other studies have shown apoptosis of MCF-7 cells following treatment with *Vitis vinifera* extracts [52]. White grape extract suppressed apoptosis in cardiomyocytes but displayed a pro-apoptotic function in MCF-7 by disrupting gap junction intracellular communication [53]. A study by Nirmala et. al. showed that in vitro treatment with *Vitis vinifera* seed and peel extracts prevented the proliferation of A431 skin cancer cells by promoting cytotoxicity, creating reactive oxygen species (ROS) accompanied by loss of mitochondrial membrane potential, and stimulated apoptosis by demonstrating morphological changes, while it was found to be non-toxic in normal human epidermal keratinocytes (HaCaΤ) cells [54]. They also showed that the inhibitory concentration (IC50), of grape seed extract was lower compared to grape peel extract (111.11 mg/mL vs. 319.14 mg/mL) in cancer cells. 

We show that treatment with *Vitis vinifera* seeds extract reduces the mRNA and protein levels of antiapoptotic proteins survivin, Bcl-2, and Bcl-xl in MCF-7 cells (Figure 4C,D). The seeds of *Vitis vinifera* are particularly rich in proanthocyanidins that can reduce the levels of Bcl-2 in cancer cells in vitro [55,56]. Trans-resveratrol obtained from *Vitis vinifera* induced apoptosis through downregulation of Bcl-2, while synthetic resveratrol is known to downregulate Bcl-xl in MCF-7 cells, causing apoptosis and growth suppression [57,58]. Grape seed proanthocyanidins reduced the expression of survivin in HepG2 cells and inhibited xenograft tumor growth in vivo [59]. Antiapoptotic proteins Bcl-2, Bcl-xl, and survivin have all been found to be affected by estrogen signaling [60,61,62]. In this study, we have seen that the *Vitis vinifera* seed extract has antiproliferative and apoptotic effect preferentially in ER+ cells; it is, therefore, important for future studies to evaluate the dependency of these extracts and their constituents on the presence of the ER. 

In this work, we did not perform purification of active constituents of *Vitis vinifera* extracts. Future research should also focus both on the sub-fractionation of crude *Vitis vinifera* extracts as well as on potential applications for their targeted delivery to cancer cells using nanoparticle formulations. *Vitis vinifera* peel and seed gold nanoparticles have already been shown to significantly reduce the number of tumors on the skin of mice when applied topically. This was attributed to the ability of the nanoparticles containing grape extract, able to increase the antioxidant enzyme activities in cells and to downregulate the expression of p53 and Bcl-2, blocking abnormal cell proliferation and inducing apoptosis [63]. In our results, we also observed Bcl-2 downregulation by *Vitis vinifera* seeds extract; in addition, p53 is wild-type in MCF-7 cells [64], which may be involved in the extract’s mode of action. *Vitis vinifera* L. seed aqueous extract was also biosynthesized in silver nanoparticles and displayed antiproliferative effects in colon cancer HT-29 cells, inducing caspase-3 cleaving and increasing the levels of p53. Interestingly, the combination of the *Vitis* extract with the common chemotherapeutic 5-FU showed synergistic cytotoxic, antiproliferative, apoptotic, and oxidative effects in vitro, suggesting a value for nanoparticle formulations of natural extracts with conventional drugs [65]. 

## 5. Conclusions

In conclusion, we measured the TPC and evaluated the antioxidant activity of twelve *Vitis vinifera* extracts. We found that the extracts displayed chemopreventive activity (as shown by their antioxidant capacity) in normal breast cells. The whole seeds, pre-washed, organic seeds ethanolic dry-extract from white grape (E7) also displayed anticancer activity in MCF-7 breast cancer cells, inducing apoptosis and decreasing the expression of survival proteins. E7 was selectively effective in cells expressing the ER, suggesting a possible role in its mode of action. Future studies should focus on determining the bioactive constituents of *Vitis vinifera* extracts, their formulation in nanoparticle carriers in combination with other agents, as well as deciphering their mode of action and potential effectiveness in vivo.

## Figures and Tables

**Figure 1 life-14-00228-f001:**
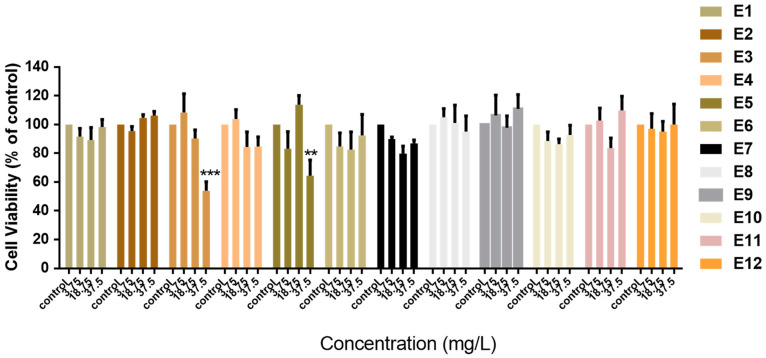
Antiproliferative effect of extracts produced from different solvents and parts of the *Vitis vinifera* plant. The cytotoxic effects of increasing concentrations of the extracts were evaluated with the MTT assay in normal MCF-10A breast cells following 72 h of incubation. All data are presented as mean values ± standard deviation and are representative of at least three independent experiments. *p* values: ** <0.01, *** <0.001, compared to control.

**Figure 2 life-14-00228-f002:**
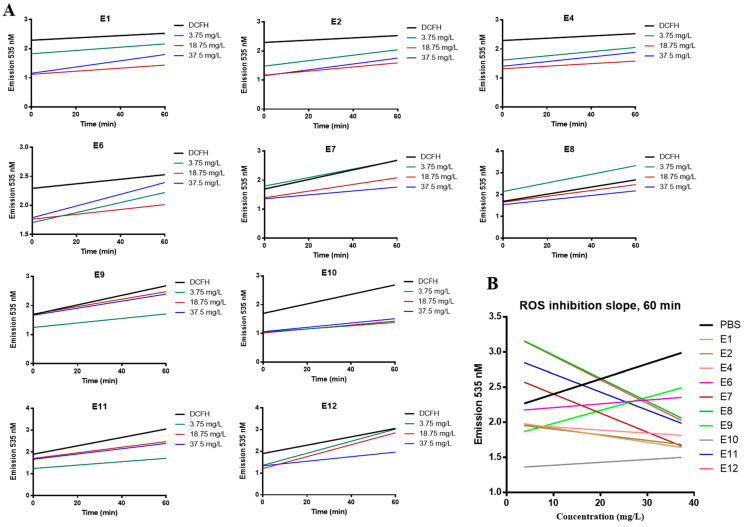
Antioxidant activity of *Vitis vinifera* extracts based on the DCFH-DA assay in MCF-10A cells. Cells were treated with 3.75, 18.75, and 37.5 mg/L of extracts in the presence of DCFH-DA and emission was measured after 0-, 10-, 20-, and 60-min incubation at 535 nm. (**A**) Time-dependent ROS inhibition per extract concentration, (**B**) dose-dependent ROS inhibition measured at 60 min for all extracts. All data are representative of at least three independent experiments.

**Figure 3 life-14-00228-f003:**
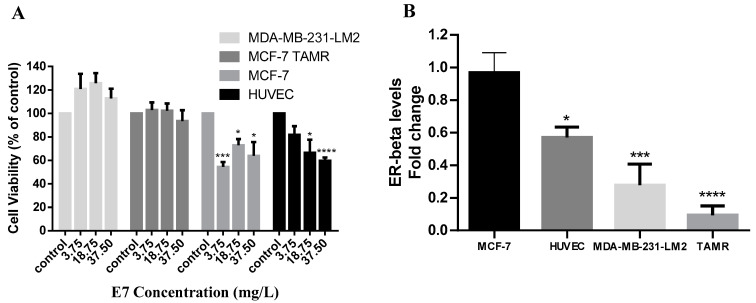
Effect of organic seeds extract E7 on breast cancer and normal cell lines and their estrogen receptor expression levels. (**A**) MCF7, HUVEC, MCF-7 TAMR, and MDA-MB-231-LM2 cells were treated with 3.75, 18.75, and 37.5 mg/L of the E7 extract and cell viability was measured by the MTT assay after 72 h of incubation, (**B**) ER-beta mRNA levels were measured using real-time PCR. All data are representative of at least three independent experiments. *p* values: * <0.05, *** <0.001, **** <0.0001 compared to control.

**Figure 4 life-14-00228-f004:**
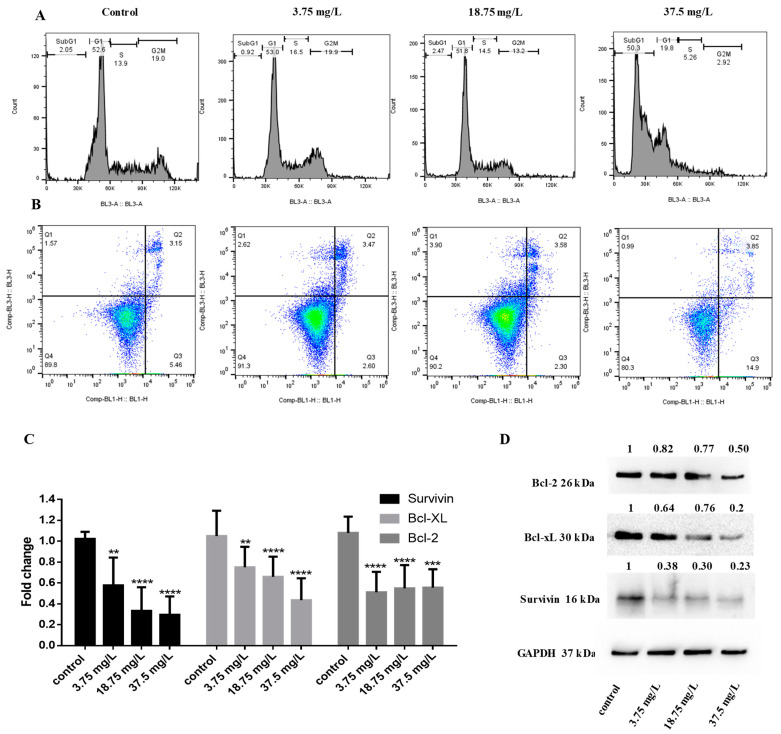
Anticancer activity of *Vitis vinifera* organic seeds extract from white grape. MCF-7 breast cancer cells were treated with 3.75, 18.75, and 37.5 mg/L of extract (**A**) Cell cycle profile analysis was performed at 48 h following incubation, (**B**) annexin V/PI staining was measured by flow cytometry after 72 h of incubation and (**C**) gene expression was analyzed using real-time PCR and (**D**) Western blot at 48 h. All data are representative of at least three independent experiments. *p* values: ** <0.01, *** <0.001, **** <0.0001 compared to control.

**Table 1 life-14-00228-t001:** Composition of *Vitis vinifera* Extracts.

Extract	Description	OPC DP 1–8 g/100 g	Catechins DP 1 g/100 g	OPC DP 2–4 g/100 g	OPC DP 5–8 g/100 g	TPC as Pyrogallol Solvent H_2_O g/100 g	TPC as Catechin Solvent H_2_O g/100 g
E1	Whole seeds dry-extract, extraction solvent EtOH 70%m/m	19.79	6.72	6.50	6.57	28.89	38.92
E2	Whole seeds dry-extract, extraction solvent H_2_O	15.48	4.74	5.84	4.91	23.53	31.74
E3	Crushed seeds without oil extract dried precipitate extraction solvent EtOH 62%m/m	9.89	2.47	3.56	3.86	12.44	17.53
E4	Crushed seeds without oil dry-extract after precipitation, dried liquid extraction solvent EtOH 62%m/m	22.37	4.89	7.81	9.68	25.94	34.33
E5	Whole seeds, organic seeds from white grape dry-extract extraction solvent EtOH 62%m/m	15.14	3.28	5.84	6.01	20.44	27.53
E6	Whole seeds, pre-washed, organic seeds, from white grape dried wash-water	7.25	1.99	2.92	2.35	8.36	11.25
E7	Whole seeds, pre-washed, organic seeds from white grape dry-extract extraction solvent EtOH 62%m/m	16.20	3.11	6.13	6.97	25.00	33.02
E8	Fruits without seeds dry-extract extraction solvent EtOH 62%m/m	0.44	0.38	0.06	<0.005	3.82	5.50
E9	Fruits without seeds prewashed, dried wash-water	0.13	0.13	<0.005	<0.005	1.47	2.39
E10	Fruits without seeds pre-washed dry-extract extraction solvent EtOH 62%m/m	0.93	0.55	0.31	0.08	6.90	9.33
E11	Leaves dry-extract extraction solvent EtOH 62%m/m	0.49	0.47	0.01	<0.005	2.97	4.74
E12	Wood dry-extract extraction solvent EtOH 62%m/m	0.84	0.77	0.07	0.01	0.86	1.66

**Table 2 life-14-00228-t002:** Antioxidant activity of *Vitis vinifera* extracts in MCF-10A cells.

Extract	Slope +	CAA50 * (mg/L)
E1	−0.0100 ± 0.007	31.00
E2	−0.0080 ± 0.007	38.94
E3	Not tested **	Not tested **
E4	−0.0043 ± 0.006	36.24
E5	Not tested **	Not tested **
E6	0.0053 ± 0.006	Not evaluable ***
E7	−0.0271 ± 0.007	53.82
E8	−0.0326 ± 0.012	57.82
E9	0.0185 ± 0.008	Not evaluable ***
E10	0.0040 ± 0.006	Not evaluable ***
E11	−0.0256 ± 0.008	54.46
E12	−0.0335 ± 0.020	66.33

+ Slope of ROS inhibition graphs at 60 min. * Concentration of extract to achieve 50% of cellular antioxidant activity (CAA). ** Extracts that were cytotoxic against MCF10A cells were not tested for antioxidant activity. *** The concentration is marked as not evaluable, in the cases where CAA did not reach 50%.

## Data Availability

The data presented in this study are available within the article and Appendix A.

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
