# Peer review of "Antioxidant and Anticancer Activity of Vitis vinifera Extracts in Breast Cell Lines"

_life, 2024, doi:10.3390/life14020228_

Round 1

Reviewer 1 Report

Comments and Suggestions for Authors

In the presented article “Antioxidant and Anticancer Activity of Vitis Vinifera Extracts in Breast Cell Lines” Evgenia Maria Tsantila et al. conducted a comparison of extracts obtained from white grapes by level of the antioxidant activity and the effect on healthy and cancer breast cells in cell cultures.

The work is good planned, the design of the study allows for authors to achieve a solution of the posed questions  and allows  to some extent evaluate the possible mechanisms of action of the extracts.

There are no fundamental comments for the materials presented in the article.

At the same time, there are a number of issues that need to be further discussed in the article.

1. Why extracts were isolated from white grape components. Numerous studies have shown that red grapes contain more polyphenols than white ones. It's need to disscused.

2. In the conclusion of the article, the sentence “This extract was” suddenly appears selectively effective in cells expressing the ER, suggesting a possible role in its mode of action". What extract are your talking about? Apparently E-7. Need to clarify

3. I would recommend that authors include additional links in the article, which shows effetivnes of grape compounds in metabolic syndrome and rheabilitation technologies.

Kubyshkin, A., Ogai, Y., Fomochkina, I., Chernousova, I., Zaitsev, G., & Shramko, Y. (2018). Polyphenols of red grape wines and alcohol-free food concentrates in rehabilitation technologies (pp. 99-120). InTechOpen.

Kubyshkin, A., Shevandova, A., Petrenko, V., Fomochkina, I., Sorokina, L., Kucherenko, A., ... & Fomochkin, I. (2020). Anti-inflammatory and antidiabetic effects of grape-derived stilbene concentrate in the experimental metabolic syndrome. Journal of Diabetes & Metabolic Disorders, 19(2), 1205-1214.

4. It would be good to conduct a correlation analysis between the antioxidant activity of the extracts and their effect on normal and cancerous breast cells.

5. The work lacks of the composition of the extracts, at least for the main components, which are then discussed in the work: catechin, epicatechin, flavanols, resveratrol, proanthocyanidins e.t.c.

Comments on the Quality of English Language

No

Reviewer 2 Report

Comments and Suggestions for Authors

The antioxidant and anticancer activities of different grape extracts on mammary gland cells were investigated. However, the manuscript readability could be much improved to better convey the importance of your study. With editing and some revisions, I feel that this manuscript will be suitable for publication.The significant weaknesses that need to be satisfactorily addressed as listed below.

1. In the extraction method, there is no clear description of the various variables and the relationship between them, and whether different grape varieties will have an impact under the same extraction method.An important issue is insufficient validation and lack of evidence to support whether there is an interaction between variables.

2. The white grape extract expressed in this paper should be used uniformly in the article, but some of them are expressed as grape extract and should be corrected in the article.

3. Line 58  Grape seeds have antioxidant activity, while leaf extract has a protective effect on oxidant ROS. The two conclusions are contrary, and data should be consulted to check whether leaf extract also has antioxidant activity.

4. Line 66  The antioxidant properties of grape extract depend on the source of the material, including the leaves, suggesting that the extract in the leaves has antioxidant activity, contrary to the conclusion above that the leaf extract has a protective effect against oxidant ROS and should be modified.

5. Line 241  It is not specified which diagram is of cells treated with extracts of different concentrations and should be added.

6. Line 576  The result of the downregulation of Bcl-xl in MCF-7 by synthetic resveratrol is not explained and should be added.

7. Line 583  Why focus on crude grape extracts? What is crude grape extract that should be given a detailed description.

8. Line 588  The nanoparticles should be replaced with nanoparticles containing grape extract.

Reviewer 3 Report

Comments and Suggestions for Authors

The manuscript presented the isolated Vitis Vinifera extracts using different methods and evaluated their anti-oxidant and anti-cancer activitites on several breast cancer cells. In my opinion, I believed the manuscript can be accepted if authors added some data to make it more readable to readers. The following listed a part but not the whole.

1. I suggeted authors to determine the possible chemical contents of extracts E1-E4, particularly E7, using LC-MS/MS and NMR methods in order to make clear which compound(s) might largely contribute to the anti-oxidant and anti-cancer activitites.

2. Authors should give the reasons why they use the same concetrations (3.75, 18.75 and 37.5 mg/L) to evaluate two totally different biological activities?

3. I do not think the anticancer 

4. Other questions. Figure 1 is meaningless which should be deleted. The concentration expression (mg/L)should be replaced by mol/L. Delete the table in Figure 2B.

Comments on the Quality of English Language

Quality of English Language can be improved.

Reviewer 4 Report

Comments and Suggestions for Authors

The manuscript: Antioxidant and Anticancer Activity of Vitis Vinifera Extracts in Breast Cell Lines

Journal: Life

Section: Pharmaceutical science

Special issue: Exploring the Potential of Natural Compounds as Anticancer Agents

Comments to the authors

Line 55 – please rephrase the sentence quoted from Sousa et al. Arrowroot is not related to species Vitis vinifera

Line 59 – recent evidence instead recent evident

The sentences from line 71 to line 81 are more appropriate for Conclusion part.

Please describe more clearly the process of seeds production. There is no data on production of plant material used for extracts E6, E7, and E11.

Please clarify and describe in detail the TPC method. In the Pharmacopoeial method there is a formula that cannot be applied to your procedure. Have you measured the reference substances in different concentrations to establish the linear regression? If yes, please add equation for pirogallol and catechin.

The OPC content method should be described in more detail.  In Table 1 you gave results for OPC DP 2-4, OPC DP 5-8 and DPC DP 1-8. What procedure did you use to differentiate the mentioned OPC groups?

Lines 195-197 need to be aligned.

In MTT assay procedure you have written that the media is removed after incubation with MTT and DMSO is added. Why did you remove the media with MTT when the principle of the method is conversion of MTT to purple formazan?

Line 308 – why did you use methanol as a solvent only for extract E3? The rest of the samples would probably show better solubility in MeOH than in water and the results would be different.

Figure 2 must be rearranged due to overlapping with the line numbers.

Figure 3 – in DCFH-DA assay was described that ˝DCHF is oxidized later to form highly fluorescent DCF, which is measured at excitation 485 nm/emission 535 nm˝. But in the Figure 3 you have described levels of DCFH as fluorescent signal. Also, for E8, Fig 3A, we can see that at the concentration of 3.75 mg/ml you have increased signal compared to DCFH and in the line 353 there is a sentence ˝E7, E8 and

E12 showed a dose-depended inhibition of ROS˝. Please explain.

You have chosen to further investigate only E7 due to best correlation between the dose and the ROS inhibition at all time points. How du you comment CAA50 of extracts E1,E2 and E4 in comparison to E7 and why these extracts were not good candidates for further antiproliferative assays?

Since E7 is the only extract tested on breast cancer cells I recommend that you revise the title which implies that all extracts have been tested for their anticancer activity.

In the discussion part you have concluded that ˝… we found that extracts E3 and E5 that had high TPC content (both measured in Pyrogallol and Catechin), were also cytotoxic against normal MCF-10A 523 breast cells (Fig. 2A, B). However, there was no direct correlation between the TPC content and cytotoxicity suggesting that other extracts constituents are responsible for their antiproliferative effects˝ and ˝Isolated compounds from Vitis Vinifera, including stilbenes, have shown selective cytotoxicity against cancer cells but not normal human fibroblasts˝.

These conclusions point to the necessity of detailed chemical characterization using liquid chromatography,  for at least extract E7.

With what results can you justify the sentence ˝We found that the extracts displayed chemopreventive activity in normal breast cells˝. Do you refer to antioxidant activity?

Comments on the Quality of English Language

Minor editing i required

Round 2

Reviewer 2 Report

Comments and Suggestions for Authors

Thank you very much!

Author Response

We thank the Reviewer for the kind comments.

Reviewer 3 Report

Comments and Suggestions for Authors

Accept in present form

Comments on the Quality of English Language

Quality of English can be improved with minor editing

Author Response

(The authors gave the same response as above.)

Reviewer 4 Report

Comments and Suggestions for Authors

Please add reference for the MTT test

Author Response

The reference for the MTT test has been added (ref 30)